# From amino acid mixtures to peptides in liquid sulphur dioxide on early Earth

Fabian Sauer [1], Maren Haas [1,2], Constanze Sydow [1], Alexander F. Siegle [1], Christoph A. Lauer [1] & Oliver Trapp [1,2 ✉]

The formation of peptide bonds is one of the most important biochemical reaction steps. Without the development of structurally and catalytically active polymers, there would be no life on our planet. However, the formation of large, complex oligomer systems is prevented by the high thermodynamic barrier of peptide condensation in aqueous solution. Liquid sulphur dioxide proves to be a superior alternative for copper-catalyzed peptide condensations. Compared to water, amino acids are activated in sulphur dioxide, leading to the incorporation of all 20 proteinogenic amino acids into proteins. Strikingly, even extremely low initial reactant concentrations of only 50 mM are sufficient for extensive peptide formation, yielding up to 2.9% of dialanine in 7 days. The reactions carried out at room temperature and the successful use of the Hadean mineral covellite (CuS) as a catalyst, suggest a volcanic environment for the formation of the peptide world on early Earth.

[1] Department of Chemistry and Pharmacy, Ludwig-Maximilians-University, Butenandtstr. 5-13, 81377 Munich, Germany. [2] Max-Planck-Institute for Astronomy, Königstuhl 17, 69117 Heidelberg, Germany. ✉email: oliver.trapp@cup.uni-muenchen.de

Because of the great variety of functions peptides can perform in nature, the formation of peptides from amino acids is irreplaceable for life as we know it[1]. This reaction plays a key role in the cellular function as well as in the origin of life. Several likely scenarios for prebiotic amino acid syntheses have been proposed[2–5]. In addition, amino acids have been identified on meteorites[6–9]. The original peptide synthesis from prebiotic material is being intensively researched today and numerous possible synthetic pathways have been proposed. The formation of peptides from amino acids is not readily performed in aqueous solution because condensation reactions are thermodynamically unfavourable under these conditions. Proposed scenarios to overcome this barrier include drying of amino acid mixtures[10,11], mineral surface catalysis[12–14], co-condensation with α-hydroxy acids[15,16], polymerization of activated amino acid derivatives[17–19], and solvent-free mechanochemical environments[20]. In salt induced peptide formation (SIPF), highly concentrated saline solutions provide dehydrating conditions, while additional metal ions, preferably copper(II), further enhance the kinetics of peptide formation[21,22]. Recent studies indicate that the salt concentration, altered by wet and dry cycles[23], might have supported the formation of cell membranes leading to protocells[24]. Furthermore, the salt composition can regulate, for example, the activity of RNA such as self-replication and extension[25]. The availability of copper(II) depends on the oxidation state of the Hadean atmosphere and would only have been possible at oxygen partial pressures of $10^{-35}$ atm[26]. Several copper minerals are assumed to have been present on the early Earth, particularly sulfides such as covellite (CuS)[27].

To circumvent the thermodynamic barrier in aqueous solution, other solvents were also considered for prebiotic peptide formation. Some alternatives such as volcanic, magmatic conditions[28], formamide as solvent[29,30], or eutectic solutions have been examined in the past[31,32]. Moreover, sulphur dioxide (SO₂) has gained attention as a prebiotically available compound in recent years, as it has been released in significant amounts into the atmosphere by volcanic outgassing and is also present in large quantities in other celestial bodies[33–35].

The question about the reaction conditions at the time of the origin of life is highly complex[36] and geochemical data are not available, because almost no rock samples from the Hadean eon are available[37] and it would be highly difficult to estimate the atmospheric pressure from them. Also, about the prevailing temperature, no certain prediction can be made at the present time since it depends substantially on the composition and chemical transformations of the atmosphere. Apart from the general assumptions about the conditions during the emergence of life (~200–800 Ma), the possibility of different, location-dependent conditions can be considered. Furthermore, seasons and different microhabitats[38] similar to today's Earth are conceivable. In recent years, detailed models to simulate the change of the atmospheric and environmental conditions of the early Earth[39] were developed.

Models predict that the probability that the surface temperature of the Earth was less than 273.15 K is 67% at ~200 Ma; thus, the climate was cold because of the consumption of CO₂ by ejecta weathering[40], which could lead to the conditions of a (soft) snowball Earth. There is evidence from samples with negative carbon isotope anomalies in carbonate rocks in Namibia for the existence of a snowball Earth later in the Earth's history, in the Paleoproterozoic (2.5 Ga)[41,42]. The atmospheric pressure ranges from ~0.01 to 100 bar[43,44]. It is also assumed that a large amount of the Earth's inventory of nitrogen was in the atmosphere with a partial pressure $p(N_2)$ of ~0.8 bar[43] or even 2–3 bar[44]. SO₂ has a boiling point of −10 °C at 1 bar. The here discussed geochemical models suggest the possibility of temporal phases or local environments on the early Earth, where SO₂ existed in liquid form. In addition, high concentrations of SO₂ near volcanoes on the Hadean Earth[45,46] increased the local partial pressure of SO₂. The influence of SO₂ has been studied in various prebiotic reactions, but mostly sulfite salts in water were used instead of liquid SO₂[47–51].

Systems of higher complexity are of particular interest in the formation of peptides, as they allow the preferential formation of specific peptide sequences, functional polymers, and even their further enrichment[52]. Prebiotic amino acid syntheses also typically lead to a variety of products[3], whose conversion results in a high diversity of peptides, depending on the reaction conditions. In recent studies, however, often only a limited number of amino acids are used simultaneously in the syntheses. The use of a larger number of amino acids in the same reaction reveals many additional aspects of the synthesis method under investigation, including frequent by-products or lack of compatibility with the reaction conditions or the other amino acids. On the other hand, it also reveals the selectivity to form certain peptides whose sequence may have greater potential for functionality due to the greater likelihood of formation.

In the present work, we investigate condensed SO₂ as an alternative hygroscopic solvent compared with aqueous salt solutions for prebiotic peptide formation. Large mixtures of amino acids are reacted in both systems and the resulting product distributions are compared. It is shown that the different environments produce varying selectivity in the peptide sequences. However, peptide condensation in SO₂ proceeds under simpler conditions, at room temperature and with low amino acid and metal catalyst loadings. Di- to tetrapeptides are formed and, importantly, all 20 amino acids can be incorporated.

## Results

**Peptide formation in sulphur dioxide.** Although the advantages of pure liquid SO₂ as a solvent in organic synthesis are well known[53], its use in the search for peptide formation reactions under prebiotic conditions has not yet been considered. Starting with the simplest amino acids glycine (G) and L-alanine (A), we investigated the possibility of efficient peptide condensation in liquid SO₂ by metal catalysis. For this purpose, the two amino acids were mixed with CuCl₂ in a pressure apparatus, SO₂ was added, and the mixture was stirred for 1–21 d at room temperature. The reaction mixture was analyzed by capillary electrophoresis (CE) and high-pressure liquid chromatography (HPLC), both coupled with high-resolution mass spectrometry (HR-MS). No peptides were observed without CuCl₂ as additive. When CuCl₂ was added, glycylglycine (GG) was already formed after 1 d; alanine-containing dipeptides as well as traces of the tripeptide GGG were detected after 7 d. Longer reaction times led to a further increase in peptide products. The cyclic diketopiperazines, which are often observed as a by-product in peptide condensation reactions[54], were not detected.

The next step was to investigate the effect of additives to the reaction mixture. High amounts of NaCl enable the SIPF reaction in water and were therefore also investigated in this context. NaCl did not increase the reactivity, on the contrary, it inhibited the product formation at high salt concentrations, similar to the reaction in water.

Another additive that was investigated is urea. Its prebiotic synthesis has already been described[55,56]. Urea has already been used as a condensation agent in prebiotic phosphorylation reactions with nucleosides[57–59]. The exact mechanism of the enhancing effect is not clear, but the hydrolysis of urea and the formation of reactive intermediate species are under discussion[57]. In addition, urea is known to form strong hydrogen bonds,

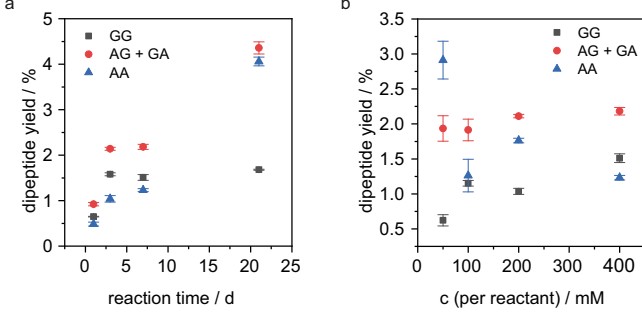

**Fig. 1 Dipeptide yields in SO₂. a** Peptide yields in dependence of the reaction time are shown with initial A, G, and CuCl₂ concentrations of 400 mM. **b** Peptide yields in dependence of the initial amino acid and CuCl₂ concentrations after 7 d are displayed. Error bars are ± s.d. after two measurements.

denature proteins, and remove water from the hydration shell of proteins[60]. A positive effect of urea on peptide condensation through the formation of reactive intermediates or as a dehydrating agent is therefore conceivable. In fact, the addition of urea to peptide formation in SO₂ not only led to an enhancement of already detected peptide signals, but it also supported the formation of the previously undetected tripeptides AAA, AAG, and AGG. The use of urea without a metal catalyst did not lead to peptide formation. Thus, urea does not catalyze the reaction itself, but promotes it as a condensation agent. For the further investigations, an equivalent of urea was added to all reactions. The peptide yields as a function of reaction time are shown in Fig. 1a. Significant amounts of dipeptides were already formed after 1 d and, except for GG, the yields increased from then until 21 d. The concentration of GG remained rather constant after 3 d, but significant amounts of tripeptides also appeared after 7 d. It is possible that GG is converted to tripeptides more rapidly than the other dipeptides.

A critical point in prebiotic reactions is often the required high concentration of reactants, which cannot be reconciled with a dilute ocean or pond on early Earth. We started the reaction with a typical concentration of 400 mM per amino acid and CuCl₂, but also tested lower concentrations of 200 mM, 100 mM, and 50 mM. Contrary to our expectations, the reaction was favoured at lower concentrations, leading to the formation of more tri- and even tetrapeptides at 50 mM after 21 d (Supplementary Figs. 41–44). Figure 1b shows the dipeptide yields after 7 d as a function of the initial amino acid and CuCl₂ concentrations. With decreasing amino acid concentration, the dipeptide yields remained constant. The maximum yield for AA was observed at an initial reactant concentration of 50 mM, suggesting effective peptide synthesis even in dilute solutions. Furthermore, significant amounts of tri- and tetrapeptides could be detected after 21 d. To further investigate the reaction in dilute solutions, we also reduced the catalyst loading. If the metal salt was not used stoichiometrically, but in 1 mol%, the reaction still led to dipeptide formation.

To further test the plausibility of the reaction under prebiotic conditions, we replaced the synthetic CuCl₂ catalyst with a natural sample of the copper(II) mineral covellite (CuS). Covellite is one of the 13 copper minerals, which may have been present in the Hadean and constituted 2.1% of the prebiotic mineral inventory[24]. It is assumed that covellite was formed by hydrothermal alteration and is one of the plausible copper minerals prior to significant near-surface oxidation[24]. With mineral catalysis in the presence of urea, dipeptides GG, AA, GA and AG were formed from 400 mM A and G after 7 d

(Supplementary Fig. 19). The amounts of dipeptides formed were about one-tenth of those obtained with synthetic CuCl₂ as catalyst. Tripeptides were not detected. Covellite thus catalyzes the reaction, despite a somewhat slower turnover, supporting the importance of this reaction scenario for the early Earth.

**Application to large amino acid mixtures.** Prebiotic peptide formation is often tested for only a limited number of amino acids. If many compounds are used, however, they are usually tested individually in separate reactions rather than in a mixture. However, these concepts overlook the dynamics and possible interactions of all amino acids with each other, which must inevitably also have taken place under the complex conditions of the early Earth. Therefore, the study of complex larger systems is becoming increasingly important. This approach allows the whole spectrum of products to be studied under specific reaction conditions and compared with others. The development of more sophisticated analytical tools allows the detailed study of the immense number of products generated during peptide condensation (peptide coupling of all proteinogenic amino acids with each other already yields 400 possible dipeptide and 8000 tripeptide sequences). The combination of efficient CE separations with fast, high-resolution mass analyzers is a powerful tool of increasing relevance and is often used for the precise analysis of protein mixtures[61]. To this end, we have recently developed a straightforward sheath-flow CE–MS interface for separation with high-sensitivity Orbitrap mass detection[62]. Tandem MS measurements (MS/MS) were performed to verify the detected peptides and determine peptide sequences.

After identifying SO₂ as an interesting alternative solvent for prebiotic peptide condensation and optimizing the reaction conditions, we extended the system to the complete set of 20 proteinogenic amino acids. In this way, the formation of any peptide in question is possible, and thus a meaningful evaluation of the system is achievable. For a conclusive interpretation of the results, we compared the dipeptide mixtures formed with those of the corresponding reaction under equivalent conditions in aqueous solution. We observed the same positive effect of urea on peptide formation in aqueous solution, so it was added to the initial SIPF conditions. Thus, the reaction time, the initial amino acid and urea concentration, and the sample concentration during analysis were identical in both solvents. However, in a few cases, the conditions differed based on previous findings: in the SIPF reaction, CuCl₂ and NaCl cannot be used in stoichiometric amounts, as there are threshold concentrations that must be present for product formation to occur at all. As shown earlier, even minute amounts of CuCl₂ induce peptide formation in SO₂ and the hygroscopic environment is sufficient to make NaCl obsolete. Therefore, fixed CuCl₂ and NaCl concentrations of 400 mM and 4.4 M were used in water, while stoichiometric amounts of CuCl₂ were used in SO₂ (1/n equivalents for mixtures of n amino acids). NaCl was not used at all. Furthermore, room temperature has been shown to be sufficient for a successful reaction in SO₂, whereas a temperature of 85 °C is used in water.

Thus, in addition to the complete set of 20 proteinogenic amino acids, four subgroups were analyzed as a function of side-chain properties. The amino acids were divided into a nonpolar, polar-neutral, alkaline, and acidic group and analyzed separately (Fig. 2). In addition, a selected mixture of amino acids that were probably first present on the early Earth[63,64] was examined. The total concentration of amino acids and urea was maintained at 400 mM in all cases and samples were taken after 7 d and 21 d.

The reaction of the nonpolar mixture in SO₂ yielded every possible dipeptide of the product spectrum (Fig. 3a). After 7 d, few valine-containing dipeptides were not yet formed, but these

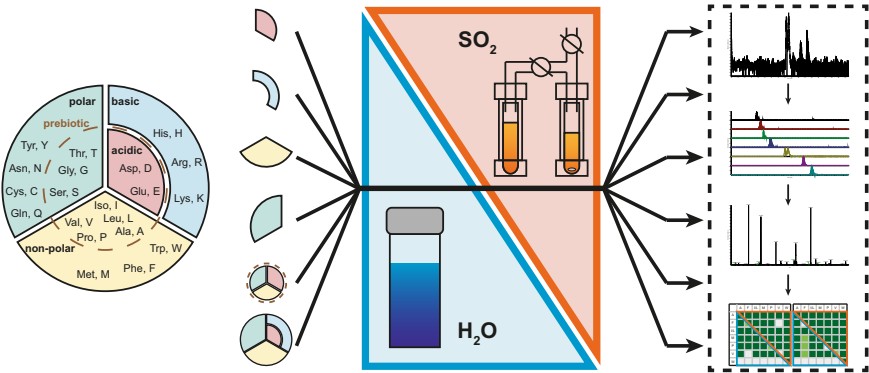

**Fig. 2 Experimental and analytical routine.** The proteinogenic amino acids were divided into subgroups based on their properties. Each of these subgroups reacted under the same reaction conditions with an overall amino acid concentration of 400 mM in $SO_2$ and water. In addition, the complete set of the proteinogenic amino acids was used in two different overall amino acid concentrations (400 mM and 50 mM). After electrophoretic separation of the reaction mixtures after 7 and 21 d, the resulting dipeptides were confirmed by tandem mass spectrometry allowing for a thorough comparison of the product distribution.

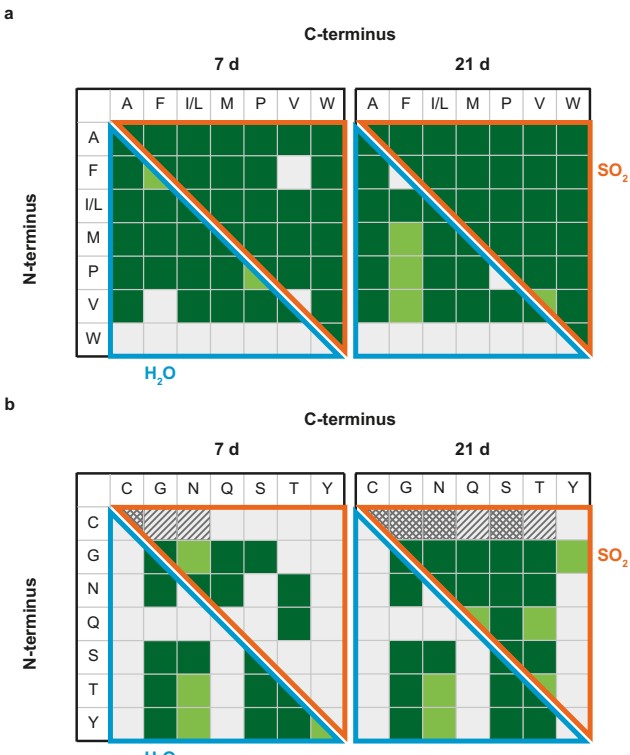

**Fig. 3 Product spectrum of the nonpolar and polar-neutral amino acid mixture.** Comparison of the detected dipeptides of copper-catalyzed peptide condensation in $SO_2$ and water after 7 and 21 d starting from the nonpolar (**a**) or polar-neutral (**b**) mixture (dark green = confirmed by MS/MS, light green = traces, white = not detected, cross-hatched = cystine peptides confirmed by MS/MS, hatched = cystine-peptide traces). Amino acids are represented corresponding to the one-letter code.

appear after 21 d. Similarly, most of the possible products are also formed in aqueous solution. However, tryptophan (W) shows low stability in solution and is susceptible to side reactions, especially in combination with metals or high temperatures[65]. After 7 d, this amino acid was rarely detected, and accordingly, dipeptide formation of W in aqueous solution occurred only very rarely. Apart from this, traces of tripeptides were detected in both reaction media.

The polar-neutral mixture was less reactive in both reaction media (Fig. 3b). In $SO_2$, tyrosine (Y) showed less reactivity, possibly due to its low solubility in the absence of acidic or basic conditions. The formation of dipeptides containing cysteine (C) could not be observed, however, its oxidation to cystine could be confirmed by MS/MS (for MS data, see Supplementary Fig. 38). Furthermore, cystine formed peptide bonds with other polar amino acids as both N- and C-terminal residues (Supplementary Fig. 39). Apart from this, all dipeptides were observed after 21 d. In aqueous solution, no cysteines could be detected either, but no disulfide bridges were formed here. Furthermore, the hydrolysis of asparagine (N) and glutamine (Q) to the respective acidic amino acids aspartic acid (D) and glutamic acid (E) led to a rapid decrease in their concentration. This strongly inhibited dipeptide formation with Q in particular. Small amounts of tripeptides could be detected in both reaction media, especially those containing two glycine molecules. Importantly, they were formed in higher amounts in $SO_2$, so that their sequence determination was possible (GGG, GGN, NGG, GGQ, QGG, GGS, GSG, SGG, GGT, GTG, and TGG).

For the alkaline and acidic mixtures, almost all possible dipeptides could be detected. The aqueous alkaline mixture showed a strong preference for the incorporation of lysine, as no other dipeptides were confirmed. No tripeptide formation was detected in either solvent (Supplementary Figs. 24–27).

In both media, numerous dipeptides were formed from the prebiotic mixture containing the amino acids proline (P), serine (S), threonine (T), leucine (L), isoleucine (I), A, D, E, G, and V, which were predominant on the early Earth (Fig. 4). The acidic amino acids were less favoured for incorporation, especially in $SO_2$. Instead, G, I, and L were the most reactive in both solvents, as well as P in $SO_2$ and V in water. The preference of one sequence over the inverted sequence was observed for several peptides (e.g., VT > TV, DG > GD). The complete dipeptide product tables with differentiation of the peptide sequences can be found in the supplementary information (Supplementary Figs. 19–36).

Finally, we performed the reaction with all 20 proteinogenic amino acids to investigate the reactivity of the overall system (Fig. 5). Some trends from the smaller subgroups remained in the overall mixture, while new observations were also made. In water, the lack of reactivity of W, Q, C, and H was still noticeable. In contrast, Y and the acidic amino acids were rather inactive in $SO_2$. Cysteine formed few dipeptides, but oxidation to cystine and further condensation with other amino acids was repeatedly observed. In both solvents, many dipeptides with amino acids I, L,

G, V, and K were detected. Other dipeptides were preferentially formed in one of the two environments (e.g., D was more reactive in water, while P and H were more reactive in $SO_2$). The composition of the amino acid mixture also affected product formation. The high activity of V in the total mixture was not predictable from the observations we made in the nonpolar mixture. In $SO_2$, V was the least reactive amino acid, while W formed fewer peptides in the total mixture. In water, far fewer dipeptides of phenylalanine (F) could be detected in the total mixture than in the subgroup. Because the amino acid concentration in all reactions was always related to the total number of amino acids, the concentration of a single amino acid in the total mixture was lower than in the other mixtures. Therefore, some dipeptides were formed in the subgroups and not in the total mixture (e.g., MI, ST, HH, and DD), and this effect was more pronounced in water than in $SO_2$. To investigate

this finding in more detail, we performed the reactions of the total mixture at a lower initial amino acid concentration (50 mM versus 400 mM total concentration, 2.5 mM versus 20 mM per amino acid). The previous experiments already showed that high yields can be obtained in $SO_2$ even at these low concentrations. However, it is only in comparison with the reaction in aqueous solution that the full potential of $SO_2$ becomes obvious (Fig. 5). In water, only very limited peptide formation is possible, resulting in remarkably few dipeptides. In $SO_2$, fewer peptides were detected, but nevertheless, peptide formation was observed for almost all amino acids, which demonstrates the effective reaction conditions impressively.

## Discussion

The formation of functional oligomer chains from single monomer building blocks remains one of the fundamental challenges that must be pursued to understand the evolution of life. In this study, we have presented an alternative reaction medium to circumvent the often discussed thermodynamically inhibited peptide condensation in aqueous environments. On the early Earth, $SO_2$ was released by volcanic emissions. As discussed in the "Introduction", geochemical models suggest the possibility of temporal phases or local environments where $SO_2$ existed as a liquid. This motivates consideration of liquid $SO_2$ as a prebiotic solvent. In our study, we were able to demonstrate effective peptide coupling by metal catalysis up to tetrapeptides starting from single amino acids.

The diverse functions that a peptide can fulfill are determined solely by its sequence. The use of many different amino acids enables the formation of an even greater variety of peptides and peptide sequences. To our knowledge, the simultaneous conversion and detailed analysis of the complete set of all proteinogenic amino acids under prebiotic conditions has not been studied before. The results of this study show that the reactivity of amino acids cannot be considered in isolation. The investigation of large mixtures reveals effects that would have been overlooked in reactions of single amino acids. In both environments studied, it was shown that the reactivity of the individual amino acid can

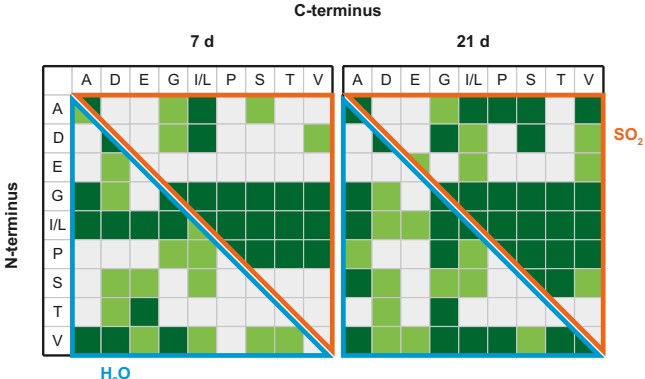

**Fig. 4 Product spectrum of the prebiotic amino acid mixture.** Comparison of the detected dipeptides of copper-catalyzed peptide condensation in $SO_2$ and water after 7 and 21 d starting from the prebiotic amino acid mixture (dark green = confirmed by MS/MS, light green = traces, white = not detected). Amino acids are represented corresponding to the one-letter code.

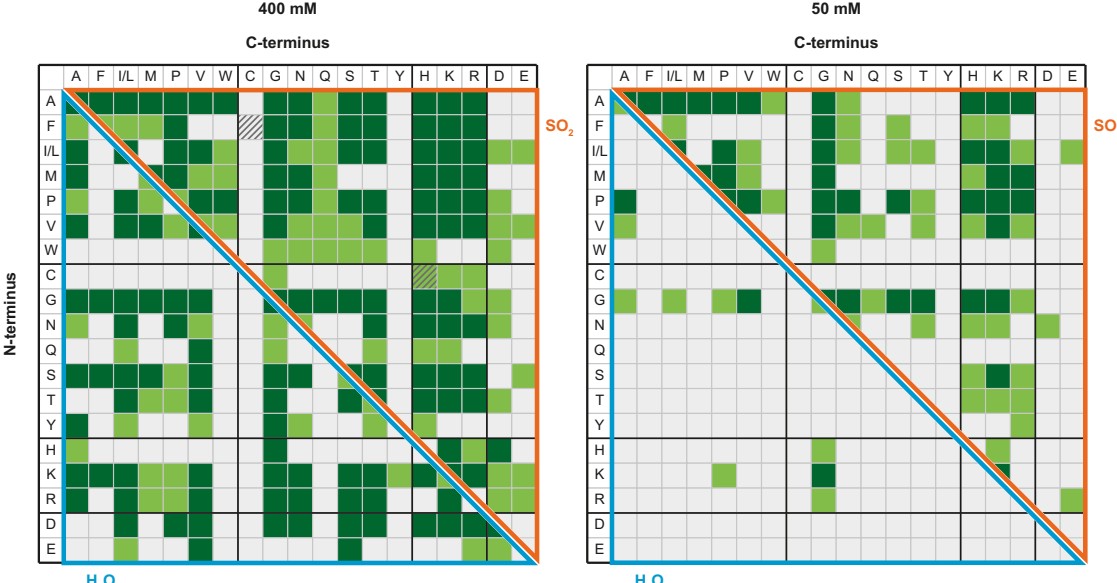

**Fig. 5 Product spectrum of the complete amino acid mixture.** Comparison of the detected dipeptides of copper-catalyzed peptide condensation in $SO_2$ and water after 21 d starting from 400 mM or 50 mM of the complete set of proteinogenic amino acids (dark green = confirmed by MS/MS, light green = traces, white = not detected, hatched = cystine-peptide traces). The $CuCl_2$ concentration in $SO_2$ was 5 mol% in both experiments. Amino acids are represented corresponding to the one-letter code.

depend on the mixture in which it is used. In SO₂, V was the least-reactive amino acid in the nonpolar mixture. In the total mixture, however, it showed increased reactivity compared with W. Furthermore, it became clear that V showed exceptionally good overall reactivity with the other amino acids. A similar observation could be made for E, which showed increased reactivity in the total mixture in SO₂ compared with the prebiotic mixture. The results show that there are cooperative effects between different amino acids, which could arise from interactions between the different side groups. For example, the hydroxyl group of S, T, or Y could form a temporary ester bond with another amino acid, thereby promoting the formation of an amide bond through an energetically favoured ester-amide exchange. This effect has already been observed in the formation of depsipeptides[15,16]. Furthermore, the catalytic activity of especially glycine and histidine in peptide formation of other amino acids has been observed in other studies[22]. In the proposed mechanism, the catalytically active amino acid promotes the formation of a mixed tripeptide and the following cleavage of it leaves a homo-dipeptide of the other amino acid.

Furthermore, a detailed comparison of the two environments investigated could be made by applying large mixtures of amino acids. The addition of urea led to increased dipeptide detection in both solvents. The comparison of the resulting dipeptide product distributions showed distinctive differences of amino acid activities in the two solvents. In water, for W, Q, C and H only a low reactivity could be observed. W, Q and N exhibit a low stability in the environment of the SIPF in which Q and N are hydrolyzed to the respective acidic amino acids. The corresponding dipeptides could only be observed in a few cases. On the other hand, the acidic amino acids D and V showed a high reactivity in water. In SO₂, rather poor reactivity of the acidic amino acids was noted. Furthermore, only traces of Y seem to be soluble in SO₂ and accordingly, only few dipeptides of those dipeptides could be detected in the reaction mixtures. However, the formation of cysteines and cystine could only be observed in SO₂. Interestingly, after oxidation to cystine, a further reaction to cystine tripeptides could be confirmed, providing sulfur bridges and the possibility for additional structural elements. Noteworthy is the increased formation of proline peptides, which are known for their catalytic activity[66]. Most importantly, each of the 20 amino acids could be incorporated into SO₂, which could not be shown for the SIPF reaction. In general, the reaction conditions in SO₂ led to a higher reactivity of the amino acids and a greater variety of products. However, the superiority of the solvent is not only reflected in the increased reactivity of the reactants but is particularly evident when compared with the corresponding reaction conditions in water, which differ in several crucial aspects. Most striking is the efficiency of peptide formation over a wide range of initial reactant concentrations. Comparatively low initial reactant concentrations of 50 mM resulted in 2.9% dialanine yield and reducing the amino acid concentration in the total mixture to even 2.5 mM still resulted in extensive dipeptide formation. Although separate environments for the emergence of the first biomolecules and the first organisms are conceivable, high metal concentrations do not seem very favorable for the first organisms. In this study, it could be shown that peptides are formed in SO₂ under strongly reduced salt and metal concentrations compared with the SIPF reaction in water. CuCl₂ could be used in catalytic amounts and other additives like NaCl were not necessary at all. Most importantly, the simple reaction conditions lead to a highly exploitative conversion at room temperature. All these findings increase the plausibility of the solvent in a prebiotic context, which is further supported by the successful product formation using the Hadean mineral covellite instead of CuCl₂. Accordingly, fewer reactants in minor concentrations at a lower temperature

are required for efficient peptide condensation in SO₂. The effectiveness of the reaction under simple conditions suggests a high potential for peptide formation in a volcanic SO₂ environment.

## Methods

**Amino acid mixtures.** For the reactions containing mixtures of amino acids, equimolar amounts of amino acids were homogenized by ball milling in a 20 mL stainless steel jar equipped with ten stainless steel balls (diameter = 10 mm) at 400 rpm for 10 min in the planetary ball mill Pulverisette 7 premium line (Fritsch GmbH, Idar-Oberstein, Germany). The detailed list of sample weights can be found in the Supplementary Information.

**Peptide condensation reactions.** Peptide formation in aqueous solution was carried out in 4 mL glass vials with magnetic stir bar. To the respective amino acid mixture (1.20 mmol in total, 1.00 eq) urea (72.1 mg, 1.20 mmol, 1.00 eq), and aqueous stock solutions of NaCl (2.64 mL, 5.00 M) and CuCl₂ (360 µL, 3.33 M) were added. The vials were sealed and placed in a metal block at 85 °C. The reactions were stirred for 21 d and samples were taken after 7 and 21 d. All experiments were performed three times.

Reactions in SO₂ (N38, 99.98%, Air Liquide Germany) were performed in stainless steel pressure apparatuses (Supplementary Fig. 1). A 3.5 mL test tube equipped with a magnetic stir bar was filled with glycine (90.1 mg, 1.20 mmol, 1.00 eq.), ʟ-alanine (106.9 mg, 1.20 mmol, 1.00 eq.), urea (72.1 mg, 1.20 mmol, 1.00 eq.), and CuCl₂ (161.3 mg, 1.20 mmol, 1.00 eq.) or optionally covellite (114.7 mg, 20.9 mmol/g). For reactions with large mixtures, the particular amino acid mixture consisting of $n$ amino acids (1.20 mmol in total, 1.00 eq.), urea (72.1 mg, 1.20 mmol, 1.00 eq.), and CuCl₂ (1/$n$ eq.) was added to the test tube instead. The test tube was inserted into the pressure apparatus, evacuated, and refilled with nitrogen three times. At −76 °C, 3 mL of SO₂ was condensed into the reaction chamber. The valves were closed, and the reaction carried out at room temperature under stirring for 1–21 d. Subsequently, the SO₂ was condensed into the storage chamber and the remaining solid dried *in vacuo*. The solvent was reused up to five times for further reactions. The test tube was removed from the apparatus and the product mixture stored at −18 °C until analysis.

**CE–MS/MS analysis.** Before measurement, the samples were diluted according to the initial amino acid concentration. In all cases, the concentration of each amino acid of the mixture in the sample vial was 1 mM. The product mixtures were analyzed using an Agilent 7100 CE system coupled to a Thermo Scientific Q Exactive Plus mass spectrometer with a custom-made sheath-flow interface, which was described in a previous study[62]. The electrophoretic separations were performed in positive polarity mode at 25 °C with an aqueous acetic acid solution (2 M) as background electrolyte (BGE). To avoid peptide adsorption linear polyacrylamide-coated capillaries with a total length of 80 cm were used whose preparation was described elsewhere[67]. Before their first utilization, a short piece of the outer polyimide coating was removed at the MS end of the capillaries and they were conditioned with deionized water (2 min), aqueous H₃PO₄ (10 mM, 5 min), deionized water (2 min), and BGE (2 min). Between measurements the capillaries were flushed again with aqueous H₃PO₄ (10 mM, 30 s), deionized water (1 min), and BGE (2 min). Sample injection was performed by applying 30 mbar pressure for 10 s. To separate the peptide mixture, a voltage of 30 kV and a constant assisting pressure of 30 mbar was applied to the CE inlet. For establishing a stable electrospray, an external voltage of 3.2 kV was applied to the stainless steel emitter while the sheath liquid was delivered at 3 µL/min. The sheath liquid consisted of deionized water and isopropanol (50:50) with 0.05% formic acid.

Mass spectra were recorded in positive mode with a resolution of 70,000 in the mass range $m/z$ 122–750. The temperature of the ion transfer capillary was set to 140 °C while a minimal flow of sweep gas was applied. The S-lens RF level was adjusted to 50. For dipeptide fragmentation, data-dependent MS/MS measurements with inclusion lists containing all possible peptide products were performed. The MS/MS spectra were measured with a resolution of 17,500 using a normalized collision energy of 30%. The mass spectra were evaluated using Thermo Xcalibur software 4.1.

**CE analysis.** Dipeptide quantification of the alanine–glycine system was performed using the Agilent 7100 CE system. Before measurement, samples were diluted to an amino acid concentration of 10 mM. The analysis was conducted in positive polarity mode at 25 °C using a conductivity detector. Separations were accomplished on bare fused silica capillaries with a total length of 80 cm and aqueous acetic acid (2 M) as BGE by applying a voltage of 30 kV. Samples were injected by applying 30 mbar for 10 s. Calibration curves of GG, AG+GA, and AA were recorded in triplicates and 4-hydroxyproline (100 µM) was used as internal standard. New capillaries were conditioned with deionized water, aqueous NaOH (0.1 M), deionized water, and BGE (each 5 min). Between measurements, capillaries were flushed with aqueous NaOH (0.1 M), deionized water, and BGE (each 2 min). CE electropherograms were evaluated using CEval 0.6g[68] and OriginPro 2018G (see Supplementary Information, Section 2).

## Data availability

The authors declare that the main data supporting the findings of this study are available within the paper and its Supplementary Information files. Extra data are available from the corresponding author upon request.

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

## Acknowledgements
We acknowledge financial support from the Ludwig-Maximilians-University Munich, the Max-Planck-Society (Max-Planck-Fellow Research Group Origins of Life), the Volkswagen Stiftung (Initiating Molecular Life), the Deutsche Forschungsgemeinschaft DFG/German Research Foundation (Project-ID 364653263—TRR 235, Emergence of Life) and Germany's Excellence Strategy (ORIGINS, EXC-2094—390783311).

## Author contributions
O.T. conceived the idea and directed the project. O.T., F.S., and M.H. designed the experiments. F.S and C.A.L. performed the experiments in aqueous salt solution. M.H. and C.S. carried out the reactions in SO₂. F.S. and M.H. developed and performed the analytical separation and identification methods. C.S., F.S., and A.F.S. validated and performed the quantification. F.S., M.H., C.S., and A.F.S. evaluated the data. M.H., F.S., and O.T. wrote the paper. All authors discussed the results and edited the paper.

## Funding

## Competing interests
The authors declare no competing interests.

## Additional information

**Peer-review information** *Nature Communications* thanks Leroy Cronin and the other anonymous reviewers for their contribution to the peer review of this work. Peer-reviewer reports are available.

