## [Peer Review File · Nature Communications]

From amino acid mixtures to peptides in liquid sulphur dioxide on early EarthREVIEWER COMMENTS

Reviewer #1 (Remarks to the Author):

This work is a very nice result in that they can see dipeptides in sulfur dioxide in most combinations of AAs and at lower concentration compared to the equivalent reaction in water. I think this lays a really important new foundation in the formation of peptides.

However, there is a lot of discussion here relating to sequence and function, while the longest they see is trace tetramers in limited cases, so it's probably a bit oversold -> I think they need to frame this as a neat (prebiotic potentially) way of making peptide /bonds/ in an anhydrous environments.

Having said that, the work is of significance and very high quality and hence should be published in NCOMMS with suitable reframing.

Notes:

The authors talk about complexity and minerals in polymerisation but do not refer to other leading work in the area of complex mixtures which seems to have been left out in favour of older studies that may not be as relevant. See: *Angew. Chem. Int. Ed.*, 2019, 58, 11253-11256 and *Proc. Natl. Acad. Sci. USA*, 2019, 116, 5387-5392 - the context in this work might help reframing here.

The authors show that when mixtures of amino acids are reacted under SPIF conditions with sulfur dioxide as the solvent, dipeptide formation can be observed for a large proportion of the possible amino acid combinations, even at lower concentrations, where practically none are observed in the equivalent reaction in water. This is certainly a nice result, but I have a number of concerns about the manuscript and the framing of the result as something prebiotically useful..

In the discussion, the authors note "formation of functional oligomer chains from single monomer building blocks remains a fundamental challenge" (in many Origin of Life scenarios), and I completely agree. However, the formation of dipeptides does not actually progress us very much further along that road unless larger aggregates can be formed. It is known that longer oligopeptides (certainly 5-, 6- mer and longer) can be formed in dehydrating / SPIF conditions, with the obvious possibility of achieving more complex products and approaching the size where sequences could impart real function. I assume the tri- / tetra- peptides formed here were in very low concentration, given that there is so little discussion of them and their sequences (just the short paragraph at line 187)? The authors note that they have developed an analysis method capable of separation and sequencing down to quite low concentrations of products, but do not really use it to its full advantage here.

The expression 'complex mixture' of amino acids is used several times, but I do not understand in what way the mixtures of amino acids used is 'complex', nor can I see any analysis here of how the sequences produced under different conditions might differ from one another, which could be of interest.

There is some discussion of differences in reactivity observed in the amino acids when mixed and in the different solvents towards the end of the discussion. This is interesting and could (should) be made more of in a revised manuscript.

Note: the title of the SI is not the same as the manuscript. The version on the main manuscript is better, the products are not 'complex'.

Reviewer #2 (Remarks to the Author):

In the paper by Sauer F et al, the authors have investigated the metal-catalyzed synthesis of short peptide oligomers in sulphur dioxide as an alternative prebiotic solvent for supporting condensation-dehydration reactions between amino acid monomers. The results show that only very short peptides are formed, and with a relatively low yield (few percent) compared to several other reported strategies. However, it seems that the authors have developed a robust and elegant experimental framework that allows them to investigate and characterize reactions involving a very complex mixture (all 20 proteinogenic amino acids). Hence, if several key points will be addressed, I believe that this paper is worthy of publication in Nature Communications.

Major points:

1. In section 3.1 of the Results, are the authors sure that all products are accounted for when measuring the concentration of short peptides (i.e. by also integrating the remaining monomers)?
2. In the 4th paragraph in section 3.1. in Results, it is stated that "Contrary to our expectations, the reaction was favored at lower concentrations, leading to the formation of more tri- and even tetrapeptides at 50 mM after 21 d". However, it is not clear how the authors reached this conclusion. The only quantification observed from Fig. 1 and the relevant SI figures (not referred to in the paper) are of dipeptides. If tripeptides were quantified, then data should be shown.
3. Can you estimate the total percent conversion of amino acid into (any) products? This number will be very useful for comparison with previously known methods.

Minor points:

1. When discussing prebiotic peptide synthesis in the Introduction, it will be good to cite Greenwald J et al. (Angewandte 2016), in which carbonyl sulfide was used to polymerize amino acids into peptides.
2. Did you check if there is any racemization during the reaction (only L-amino acids were used as monomers)?
3. Supplementary figures 2-23 are not called in the main text.
4. It will be good to add a separate SI figure with the individual standards that were detected as a reference (e.g. AA, GG), as well as those that were not detected (GG DKP).
5. The authors state that "Urea has already been used as a condensation agent in prebiotic phosphorylation reactions with nucleosides^{41,42}". Burcar B et al (Angewandte 2016) should also be added as a reference to this sentence.
4. Raw data is not shown for Supplementary Fig. 19.
5. To me, one of the very interesting findings of this paper was that the analysis of complex mixtures reveals effects that would have been overlooked in reactions of single amino acids. While this is mentioned in the Discussion, no possible explanation is offered. I believe it will be good to discuss some possible explanations.

Reviewer #3 (Remarks to the Author):

Overview

This is a creative study which explores the chemistry of amino acid oligomerization in various solvents. Notably, this study finds that liquid SO₂ promotes AA oligomerization, even at low reactant concentration, using prebiotically plausible mineral counterparts.

Major comments:

This manuscript presents compelling synthetic chemistry. However, the authors have not adequately established that this synthetic chemistry is also *prebiotic* chemistry, i.e. that it is strongly relevant to early Earth.

My strongest criticism is the plausibility of condensed (liquid) SO₂ on early Earth. The boiling point of SO₂ is 263 K at 1 atm (<https://pubchem.ncbi.nlm.nih.gov/compound/Sulfur-dioxide>). Most

models of early Earth assume ~ 1 atm surface pressure and predict mean surface temperatures above freezing in order to explain the zircon evidence. In this region of T-P space, liquid SO₂ cannot exist. I do not agree with the offhand claim of this manuscript that liquid SO₂ environments are highly *likely* on early Earth.

Liquid SO₂ may have been *possible* if early Earth did not conform to the "standard" picture, many aspects of which are only weakly justified. For example:

- A high surface pressure for early Earth is not predicted from models (e.g. Kadoya et al. 2018), but is also not ruled out. Surface pressures of up to 10 bar are proposed (Kasting et al. 1993). Such conditions might allow liquid SO₂, though I emphasize that these are extremal scenarios not consistent with current predictions of early Earth.

- A cold early Earth is possible, particularly on a transient basis (Kadoya et al. 2018). Such a planet might have hosted temperatures locally (e.g., at the poles) or globally.

The authors must identify the scenarios in which SO₂ could be stable, and discuss their plausibility/prevalence in the text. I suggest the authors plot a phase diagram of SO₂ and show on this phase diagram the different scenarios under which liquid SO₂ would have been stable.

Note I have discussed here only concerns about liquid SO₂ from the T/P perspective. There are other concerns as well. For example, SO₂ is photochemically unstable. Could pure lakes of it have accumulated? This SO₂ would also likely have been mixed with substantial amounts of water, which is also volcanically outgassed and which was also supplied by the hydrological cycle. Does their chemistry also work in mixed SO₂-H₂O solution?

A more minor concern relates to the authors' use of copper. Copper is geologically rare. The authors invoke Covellite as a plausible Cu(II) source. Can the authors comment on the estimated prevalence of covellite on early Earth?

Minor comments:

In exploring SIPF: Can you comment on the implications of high salt concentrations for other parts of the protolife apparatus, like vesicles and nucleic acids.

Lines 50-51: At face value, this sentence implies Cu(II) would not have been available on early Earth, since photochemical oxygen abundances are predicted to exceed 10^{-35} atm by many orders of magnitude (e.g., Haqq-Misra, Kasting & Lee 2011).

Line 60, 230-231: There is no consensus that early Earth had higher p than modern, and even some indications that it had lower p than modern (e.g., Som et al. 2012, 2016; Gebauer, Grenfell, Lammer. et al. 2020)

Line 62: Other important references here include Becker et al 2019 (Science), and Xu et al. 2018 (ChemComm).

Lines 65-67: I commend the authors for acknowledging the inevitable complexity and non-ideal conditions that must have been present in prebiotic chemistry.

Line 118-121: The authors have done well in considering the scaling of their reaction with reactant concentration, thus remedying a common criticism of prebiotic chemistry. However, 50 mM of amino acids are not low concentrations. I would describe that as quite concentrated, albeit less so than what is sometimes assumed.

Line 127-132: On what basis is it known that covellite was Hadean? How widespread was it on early Earth?

Lines 130-132: Can the authors please expand on the results. From what is written, it sounds like covellite is much worse than pure Cu(II) as a catalyst (e.g., no tri- or tetra-peptides). How does the yield compare.

Lines 135-148: Bravo to the authors! This is exactly the kind of work that is needed to move prebiotic chemistry from the lab to nature, i.e. convert it from synthetic to prebiotic chemistry.

Lines 197-200: What were the trends in dipeptide formation? Did they, e.g., match what is observed in biology?

Lines 260-262: please quantify effect on yields.

List of Changes

Peptide formation as on the early Earth: from amino acid mixtures to peptides in sulphur dioxide

On behalf of all the authors, I would like to thank the competent reviewers for providing us with great feedback on our manuscript. We greatly appreciate all the helpful suggestions and valuable comments provided by the reviewers to improve the quality of the manuscript.

Reviewer #1 (Remarks to the Author):

This work is a very nice result in that they can see dipeptides in sulfur dioxide in most combinations of AAs and at lower concentration compared to the equivalent reaction in water. I think this lays a really important new foundation in the formation of peptides.

However, there is a lot of discussion here relating to sequence and function, while the longest they see is trace tetramers in limited cases, so it's probably a bit oversold -> I think they need to frame this as a neat (prebiotic potentially) way of making peptide /bonds/ in an anhydrous environments.

Having said that, the work is of significance and very high quality and hence should be published in NCOMMS with suitable reframing.

- We thank the reviewer for acknowledging the great potential of SO₂ as alternative solvent for prebiotic peptide formation.

Notes:

The authors talk about complexity and minerals in polymerisation but do not refer to other leading work in the area of complex mixtures which seems to have been left out in favour of older studies that may not be as relevant. See: Angew. Chem. Int. Ed., 2019, 58, 11253-11256 and Proc. Natl. Acad. Sci. USA, 2019, 116, 5387-5392 - the context in this work might help reframing here.

- Thank you very much for bringing these publications to our attention. We have added these references.

The authors show that when mixtures of amino acids are reacted under SPIF conditions with sulfur dioxide as the solvent, dipeptide formation can be observed for a large proportion of the possible amino acid combinations, even at lower concentrations, where practically none are observed in the equivalent reaction in water. This is certainly a nice result, but I have a number of concerns about the manuscript and the framing of the result as something prebiotically useful..

- Thank you for these encouraging comments. We would like to point out that sulphur dioxide might have not only played an important role as solvent in various environments on the early Earth, but also on other planets with intense volcanic activity.

In the discussion, the authors note "formation of functional oligomer chains from single monomer building blocks remains a fundamental challenge" (in many Origin of Life scenarios), and I completely agree. However, the formation of dipeptides does not actually progress us very much further along that road unless larger aggregates can be formed. It is known that longer oligopeptides (certainly 5-, 6- mer and longer) can be formed in dehydrating / SPIF conditions, with the obvious possibility of achieving more complex products and approaching the size where sequences could impart real function. I assume the tri- / tetra- peptides formed here were in very low concentration, given that there is so little discussion of them and their sequences (just the short paragraph at line 187)? The authors note that they have developed an analysis method capable of separation and sequencing down to quite low concentrations of products, but do not really use it to its full advantage here.

- Plausible prebiotic reactions are characterized by robust reaction pathways that create a wide variety of products under simple conditions with a limited number of reactants. Our study focuses on the development of reasonable reaction conditions in an alternative, prebiotically plausible medium, where the barriers of peptide formation can be overcome. We definitely agree with the reviewer that the formation of dipeptides alone does not advance us in creating more functional peptides. The formation of longer peptides is for sure important, however, the function and capabilities of a specific peptide are also decisively determined by its sequence. Oligopeptides can lack function if they are assembled from the same amino acid. Therefore, complex amino acid mixtures were used in the study, as the successful conversion of these mixtures reinforces the prebiotic significance of the new reaction conditions. We could show that under the established simple environment every proteinogenic amino acid can be incorporated into dipeptides. We believe that this is an equally important aspect than the formation of longer peptides would be. Due to the high effort of the analysis (large amount of products), we limited ourselves to the evaluation of all 400 possible dipeptide sequences. However, tri- and tetrapeptides were also observed on several occasions (in the small system consisting of alanine and glycine – lines 107-114, 118-124 and 232 - and in each complex mixture – lines 176, 187, 252). We added new Supplementary Figures to further demonstrate the formation of higher peptides in the small system (Supplementary Fig. 41-44). However, the synthesis of longer peptide chains under the established conditions is part of current investigations.

The expression 'complex mixture' of amino acids is used several times, but I do not understand in what way the mixtures of amino acids used is 'complex', nor can I see any analysis here of how the sequences produced under different conditions might differ from one another, which could be of interest.

- We understand that the expression “complex mixture” can be misleading as we are referring to the number of different reactants and not the functionality of the group. We therefore replaced “complex” with “large”.

There is some discussion of differences in reactivity observed in the amino acids when mixed and in the different solvents towards the end of the discussion. This is interesting and could (should) be made more of in a revised manuscript.

- We thank the reviewer for the constructive suggestion and we further elaborated the different amino acid activities in the different environments in the Discussion (lines 244-253 and 260-267):

“A similar observation could be made for E which showed increased reactivity in the total mixture in SO₂ compared to the prebiotic mixture. The results show that there are cooperative effects between different amino acids which could arise from interactions between the different side groups. For example, the hydroxyl group of S, T or Y could form a temporary ester bond with another amino acid, thereby promoting the formation of an amide bond through an energetically favoured ester-amide exchange. This effect has already been observed in the formation of depsipeptides.^{1,2} Furthermore, the catalytic activity of especially glycine and histidine in peptide formation of other amino acids has been observed in other studies.³ In the proposed mechanism, the catalytically active amino acid promotes the formation of a mixed tripeptide and the following cleavage of it leaves a homo-dipeptide of the other amino acid.”

“The comparison of the resulting dipeptide product distributions showed distinctive differences of amino acid activities in the two solvents. In water, for W, Q, C and H only a low reactivity could be observed. W, Q and N exhibit a low stability in the environment of the SIPF

in which Q and N are hydrolysed to the respective acidic amino acids. The corresponding dipeptides could only be observed in a few cases. On the other hand, the acidic amino acid D and V showed a high reactivity in water. In SO₂, rather poor reactivity of the acidic amino acids was noted. Furthermore, only traces of Y seem to be soluble in SO₂ and accordingly, only few dipeptides of those dipeptides could be detected in the reaction mixtures.”

Note: the title of the SI is not the same as the manuscript. The version on the main manuscript is better, the products are not 'complex'.

- Thank you, we corrected the title of the SI.

Reviewer #2 (Remarks to the Author):

In the paper by Sauer F et al, the authors have investigated the metal-catalyzed synthesis of short peptide oligomers in sulphur dioxide as an alternative prebiotic solvent for supporting condensation-dehydration reactions between amino acid monomers. The results show that only very short peptides are formed, and with a relatively low yield (few percent) compared to several other reported strategies. However, it seems that the authors have developed a robust and elegant experimental framework that allows them to investigate and characterize reactions involving a very complex mixture (all 20 proteinogenic amino acids). Hence, if several key points will be addressed, I believe that this paper is worthy of publication in Nature Communications.

Major points:

1. In section 3.1 of the Results, are the authors sure that all products are accounted for when measuring the concentration of short peptides (i.e. by also integrating the remaining monomers)?

3. Can you estimate the total percent conversion of amino acid into (any) products? This number will be very useful for comparison with previously known methods.

- We examined our reactions for side products commonly occurring in peptide condensation reactions (DKPs, degradation products of amino acids), however, as already mentioned no such compounds were found. Furthermore, we investigated the conversion of the amino acids Ala and Gly at different reaction times in SO₂ by integrating the corresponding signals of CE measurements. The results showed a conversion of up to 10% for Ala and a 30% conversion for Gly. The conversion of Ala agrees very well with the observed dipeptide yields and shows that a targeted synthesis of dipeptides occurs during the reaction. However, the conversion of Gly is higher than the corresponding dipeptide yields would suggest. Therefore, minor side reactions seem to occur for Gly but not for Ala. Similar observations were also made in H₂O for Trp, Gln, Asn.⁴ We agree, that these findings are interesting, as they show an acceptable stability of the reactants in the established reaction conditions. However, the differing amounts of conversion of the two amino acids also show, that yields cannot be determined through the analysis of conversion rates alone. This study focuses on dipeptide formation of complex mixtures and on the yields which can be achieved in the new SO₂-environment. In our opinion, the comparison with conversion rates of different methods does not add further insight into this topic.

2. In the 4th paragraph in section 3.1. in Results, it is stated that “Contrary to our expectations, the reaction was favored at lower concentrations, leading to the formation of more tri- and even tetrapeptides at 50 mM after 21 d”. However, it is not clear how the authors reached this conclusion. The only quantification observed from Fig. 1 and the relevant SI figures (not referred to in the paper) are of dipeptides. If tripeptides were quantified, then data should be shown.

- We thank the reviewer for the constructive suggestion. Indeed, our conclusion is not made clear enough. We included additional figures in the Supplementary Information with a comparison of the respective EIEs (Supplementary Fig. 41-44).

Minor points:

1. When discussing prebiotic peptide synthesis in the Introduction, it will be good to cite Greenwald J et al. (Angewandte 2016), in which carbonyl sulfide was used to polymerize amino acids into peptides.

- Thank you very much for bringing this publication to our attention. We have added the reference.

2. *Did you check if there is any racemization during the reaction (only L-amino acids were used as monomers)?*

- Indeed, only L-amino acids were used in the reactions. So far, we didn't check for racemization, but we will extend this in further studies, which is an interesting point in the context of symmetry breaking.

3. *Supplementary figures 2-23 are not called in the main text.*

- Figures 19-36 (dipeptide product spectra of the different reactions) are referred to in line 200. Figures 2-18 show the calibration plots and the analysis runs for the quantification of dipeptide formation in SO₂. We added a corresponding reference in the methods section.

4. *It will be good to add a separate SI figure with the individual standards that were detected as a reference (e.g. AA, GG), as well as those that were not detected (GG DKP).*

- The EIEs of the reference peptides GG, AG/GA, AA, GGG and AGG are already included in the SI (Supplementary Figure 37). We further added a CE-MS run of the DKPs of GG and AA (Supplementary Figure 38).

5. *The authors state that "Urea has already been used as a condensation agent in prebiotic phosphorylation reactions with nucleosides^{41,42}". Burcar B et al (Angewandte 2016) should also be added as a reference to this sentence.*

- We included the requested reference.

4. *Raw data is not shown for Supplementary Fig. 19.*

- The raw data was already shown in Supplementary Figures 103 and 104.

5. *To me, one of the very interesting findings of this paper was that the analysis of complex mixtures reveals effects that would have been overlooked in reactions of single amino acids. While this is mentioned in the Discussion, no possible explanation is offered. I believe it will be good to discuss some possible explanations.*

- We thank the reviewer for the suggestion and added some possible explanations for the observed effects in the Discussion (lines 244-253):

"A similar observation could be made for E which showed increased reactivity in the total mixture in SO₂ compared to the prebiotic mixture. The results show that there are cooperative effects between different amino acids which could arise from interactions between the different side groups. For example, the hydroxyl group of S, T or Y could form a temporary ester bond with another amino acid, thereby promoting the formation of an amide bond through an energetically favoured ester-amide exchange. This effect has already been observed in the formation of depsipeptides.^{1,2} Furthermore, the catalytic activity of especially glycine and histidine in peptide formation of other amino acids has been observed in other studies.³ In the proposed mechanism, the catalytically active amino acid promotes the formation of a mixed tripeptide and the following cleavage of it leaves a homo-dipeptide of the other amino acid."

Reviewer #3 (Remarks to the Author):

Overview

This is a creative study which explores the chemistry of amino acid oligomerization in various solvents. Notably, this study finds that liquid SO₂ promotes AA oligomerization, even at low reactant concentration, using prebiotically plausible mineral counterparts.

Major comments:

*This manuscript presents compelling synthetic chemistry. However, the authors have not adequately established that this synthetic chemistry is also **prebiotic** chemistry, i.e. that it is strongly relevant to early Earth.*

*My strongest criticism is the plausibility of condensed (liquid) SO₂ on early Earth. The boiling point of SO₂ is 263 K at 1 atm (<https://pubchem.ncbi.nlm.nih.gov/compound/Sulfur-dioxide>). Most models of early Earth assume ~1 atm surface pressure and predict mean surface temperatures above freezing in order to explain the zircon evidence. In this region of T-P space, liquid SO₂ cannot exist. I do not agree with the offhand claim of this manuscript that liquid SO₂ environments are highly **likely** on early Earth.*

*Liquid SO₂ may have been **possible** if early Earth did not conform to the “standard” picture, many aspects of which are only weakly justified. For example:*

- A high surface pressure for early Earth is not predicted from models (e.g. Kadoya et al. 2018), but is also not ruled out. Surface pressures of up to 10 bar are proposed (Kasting et al. 1993). Such conditions might allow liquid SO₂, though I emphasize that these are extremal scenarios not consistent with current predictions of early Earth.*

- A cold early Earth is possible, particularly on a transient basis (Kadoya et al. 2018). Such a planet might have hosted temperatures locally (e.g, at the poles) or globally.*

The authors must identify the scenarios in which SO₂ could be stable, and discuss their plausibility/prevalence in the text. I suggest the authors plot a phase diagram of SO₂ and show on this phase diagram the different scenarios under which liquid SO₂ would have been stable.

Note I have discussed here only concerns about liquid SO₂ from the T/P perspective. There are other concerns as well. For example, SO₂ is photochemically unstable. Could pure lakes of it have accumulated? This SO₂ would also likely have been mixed with substantial amounts of water, which is also volcanically outgassed and which was also supplied by the hydrological cycle. Does their chemistry also work in mixed SO₂-H₂O solution?

*Line 60, 230-231: There is no consensus that early Earth had higher *p* than modern, and even some indications that it had lower *p* than modern (e.g., Som et al. 2012, 2016; Gebauer, Grenfell, Lammer. et al. 2020)*

- We agree with the reviewer - since this is the first time pure liquid SO₂ is recognized as prebiotic solvent - that the existence of it on early Earth has to be verified. Please note that at no point in the manuscript we are making offhand claims but attempted to substantiate the statements made with appropriate studies from experts in this field. However, there is no doubt that the question about the reaction conditions at the time of the origin of life is highly complex and to this day not solved. Almost no rock samples from the Hadean have survived, but even if they had, it would be difficult to determine the atmospheric pressure from them. Zahnle et al. assume very high levels of CO₂ (approximately 100 bar) and additional 2-3 bar N₂, others estimate 10-20 bar CO₂.⁵⁻⁸ Of course, the pressure decreases with the formation of carbonates, however, it does so in an unknown manner and period of time. Haqq-Misra et al assume a 1-2.8 bar surface pressure for the period of 3-4 Ga for their models.⁹ Also about the prevailing*

temperature no certain prediction can be made at the present time, since it depends substantially on the composition of the atmosphere. Liquid water deposits are considered to be ensured, however, subzero temperatures would have dominated without the presence of greenhouse gases because of the weaker sun. So there are large variations in the assumptions made about the early surface pressure and temperature, however, all of the mentioned conditions would support liquid SO_2 . We included a phase diagram of SO_2 below.¹⁰ Unfortunately, we could not find the reference of Kadoya et al. from 2018, so we assumed the reviewer meant either one from 2019.^{11,12} As the reviewer already mentioned, high pressures as well as low temperatures are not ruled out for the early Earth in these studies. Som et al. and Gebauer et al. acknowledge the difficulties in estimating past pressures and state that the results of studies vary strongly.¹³⁻¹⁵ However, the assumptions made by Som et al. refer to the time of 2.7 Ga, when life probably already existed on earth.¹⁶ Apart from the general assumptions about the conditions during the origin of life (which is not a fixed point in time but a range of 200-800 Ma), the possibility of different, location-dependent conditions must be considered. Furthermore, seasons and different microhabitats similar to today's earth are conceivable. We therefore think that there are no "standard" prebiotic reaction conditions but instead a wide range of different conditions has to be considered in order to explain the emergence of life. According to the current state of knowledge, liquid SO_2 can by no means be ruled out as an alternative solvent for prebiotic reactions since the above discussed scenarios would all allow for its existence.

- As described by Kasting et al. the photochemical degradation of SO_2 mainly takes place in the atmosphere with the formation of a sulphur layer.¹⁷ Dry and wet depositions are discussed in the same context. We could not find studies regarding the UV stability of SO_2 -lakes. However, the UV problem affects not only SO_2 but also other organic molecules.
- We did not investigate peptide formation in defined H_2O - SO_2 -mixtures. However, the SO_2 was not dried prior to usage and water originating from the condensation reactions was not separated from the reaction mixture. Consequently, all reaction mixtures contained small amounts of water.

A more minor concern relates to the authors' use of copper. Copper is geologically rare. The authors invoke Covellite as a plausible Cu(II) source. Can the authors comment on the estimated prevalence of covellite on early Earth?

Line 127-132: On what basis is it known that covellite was Hadean? How widespread was it on early Earth?

- Because of the low levels of oxygen in the atmosphere and especially in the aqueous, mineral-forming environments of the prebiotic Earth, the number of available copper minerals was far less than it is today. Hazen assumes an oxygen fugacity close to that of the hematite-magnetite buffer ($f_{O_2} \sim 10^{-72}$), which renders the formation of copper oxides, silicates, carbonates, sulfates, arsenates and phosphates impossible. However, Hazen also states that the formation of the copper-sulfur bond does not depend primarily on the fugacity of oxygen, but on that of sulfur. Since this was sufficiently high ($f_{S_2} > 10^{-44}$), sulfides and sulfosalts were available on the prebiotic Earth at any $f_{O_2} < 10^{-52}$. Therefore, the number of different copper minerals is estimated to be 13 (including covellite), which constituted 2.1% of the prebiotic mineral inventory.¹⁸ Therefore, especially local deposits of copper minerals cannot be ruled out. However, we are aware of the limited availability of copper(II) and we are working on more abundant alternatives in ongoing investigations.

Minor comments:

In exploring SIPF: Can you comment on the implications of high salt concentrations for other parts of the protolife apparatus, like vesicles and nucleic acids.

- Rode et al. regularly stated that peptides are more stable under high salt concentrations than nucleotides.¹⁹ They concluded that the RNA-world was preceded by a peptide world and that peptides took on a variety of different functions early in history. Therefore, peptides do not seem to be as restrictive with respect to reaction conditions as nucleotides or sugars. On the other hand, high metal concentrations do not seem very favorable for the first organisms. However, separate environments for the emergence of the first biomolecules and the first

organisms are conceivable. Unfortunately, no holistic scenario for the origin of life could be experimentally confirmed yet. However, in this study it could be shown that peptides are formed in SO₂ under strongly reduced salt and metal concentrations compare to the SIPP reaction in water. This fits in nicely with the point brought up by the reviewer that high salt concentrations might be an obstacle to the emergence of life.

Lines 50-51: At face value, this sentence implies Cu(II) would not have been available on early Earth, since photochemical oxygen abundances are predicted to exceed 10⁻³⁵ atm by many orders of magnitude (e.g., Haqq-Misra, Kasting & Lee 2011).

- Ochiai states that oxygen partial pressures of 10⁻³⁵ bar were sufficient for the occurrence of Cu(II), while Haqq-Misra et al. predict oxygen partial pressures of at least 10⁻¹³ bar near the earth's surface. Therefore, Cu(II) would have been available even before the appearance of photosynthesis.

Line 62: Other important references here include Becker et al 2019 (Science), and Xu et al. 2018 (ChemComm).

- We included the requested references.

Line 118-121: The authors have done well in considering the scaling of their reaction with reactant concentration, thus remedying a common criticism of prebiotic chemistry. However, 50 mM of amino acids are not low concentrations. I would describe that as quite concentrated, albeit less so than what is sometimes assumed.

- We agree with the reviewer that prebiotic reactions should not only take place with large concentrations. We are showing that in SO₂ products can be formed over a great range of initial reactant concentrations, while the obtained yields do not decline with decreasing concentration. The full potential of the established reaction conditions becomes obvious through the comparison with the corresponding conditions in water. In the total amino acid mixture, the concentration of a single amino acid is only 2.5 mM. Nevertheless, an extensive amount of dipeptides is formed under these low concentrations.

Lines 130-132: Can the authors please expand on the results. >From what is written, it sounds like covellite is much worse than pure Cu(II) as a catalyst (e.g., no tri- or tetra-peptides). How does the yield compare.

- The amount of dipeptides formed in the 7-day period using covellite as a catalyst is smaller compared to the amounts obtained with synthetic CuCl₂ in the same period of time. Therefore, the corresponding yields cannot be quantified by CE. A rough estimation can be made by comparing the respective dipeptide peak areas of the CE-MS measurements. After 7 days, yields obtained with synthetic CuCl₂ are approximately ten times as high as with covellite. Nevertheless, the successful application of the prebiotic mineral in low amounts (21 mmol amino acids per gram mineral as opposed to the often used 0.2 mmol amino acids per gram mineral)²⁰ demonstrates the great potential of the established reaction conditions in SO₂ for peptide formation. On a prebiotic timescale larger amounts of dipeptides will be formed.

Lines 65-67: I commend the authors for acknowledging the inevitable complexity and non-ideal conditions that must have been present in prebiotic chemistry.

Lines 135-148: Bravo to the authors! This is exactly the kind of work that is needed to move prebiotic chemistry from the lab to nature, i.e. convert it from synthetic to prebiotic chemistry.

- We are very happy the reviewer acknowledges one of the key features of the study and shares our opinion on the necessity of expanding prebiotic systems.

Lines 197-200: What were the trends in dipeptide formation? Did they, e.g., match what is observed in biology?

- We could observe different activities of the amino acids depending on the solvent and mixture the amino acid is used in. For example, we observed an increased formation of proline peptides in SO₂ which are known for their catalytic activity. Please also refer to the Discussion of the revised manuscript where we further elaborated these effects (lines 244-253 and 260-267):
“A similar observation could be made for E which showed increased reactivity in the total mixture in SO₂ compared to the prebiotic mixture. The results show that there are cooperative effects between different amino acids which could arise from interactions between the different side groups. For example, the hydroxyl group of S, T or Y could form a temporary ester bond with another amino acid, thereby promoting the formation of an amide bond through an energetically favoured ester-amide exchange. This effect has already been observed in the formation of depsipeptides.^{1,2} Furthermore, the catalytic activity of especially glycine and histidine in peptide formation of other amino acids has been observed in other studies.³ In the proposed mechanism, the catalytically active amino acid promotes the formation of a mixed tripeptide and the following cleavage of it leaves a homo-dipeptide of the other amino acid.”
“The comparison of the resulting dipeptide product distributions showed distinctive differences of amino acid activities in the two solvents. In water, for W, Q, C and H only a low reactivity could be observed. W, Q and N exhibit a low stability in the environment of the SIPF in which Q and N are hydrolysed to the respective acidic amino acids. The corresponding dipeptides could only be observed in a few cases. On the other hand, the acidic amino acid D and V showed a high reactivity in water. In SO₂, rather poor reactivity of the acidic amino acids was noted. Furthermore, only traces of Y seem to be soluble in SO₂ and accordingly, only few dipeptides of those dipeptides could be detected in the reaction mixtures.”

Lines 260-262: please quantify effect on yields.

- We quantified the obtained peptide products in SO₂ for a large range of initial reactant concentrations and different reaction times while using a small set of two amino acids (Fig. 1). We did not determine yields of the dipeptides produced from the complex mixtures as the many ions influence each other during mass analysis which renders the precise quantification very difficult. Instead we compared the resulting dipeptide product spectra of SO₂ to the ones obtained in H₂O (Fig. 5). Especially in the total mixture, the potential of SO₂ is on full display, since even at low concentrations a significantly larger amount of dipeptides is formed than in H₂O.

-
- ¹ Frenkel-Pinter, M. et al. Selective incorporation of proteinaceous over nonproteinaceous cationic amino acids in model prebiotic oligomerization reactions. *Proc. Natl. Acad. Sci.* **116**, 16338-16346 (2019).
- ² Doran, D., Abul-Hija, Y. M. & Cronin, L. Emergence of Function and Selection from Recursively Programmed Polymerisation Reactions in Mineral Environments. *Angew. Chem. Int. Ed.* **58**, 11253-11256 (2019).
- ³ Jakschitz, T. A. E. & Rode, B. M. Chemical evolution from simple inorganic compounds to chiral peptides. *Chem. Soc. Rev.* **41**, 5484-5489 (2012).
- ⁴ Bellmaine, S., Schnellbaecher, A. & Zimmer, A. Reactivity and degradation products of tryptophan in solution and proteins. *Free Radical Biol. Med.* **160**, 696-718 (2020).
- ⁵ Zahnle, K. J. Earth's Earliest Atmosphere. *Elements* **2**, 217-222 (2006).
- ⁶ Zahnle, K., Schaefer, L. & Fegley, B. Earth's Earliest Atmospheres. *Cold Spring Harb. Perspect. Biol.* **2**, a004895 (2010).
- ⁷ Kasting, J. F. & Ackerman, T. P. Climatic consequences of very high carbon dioxide levels in the earth's early atmosphere. *Science* **234**, 1383-1385 (1986).
- ⁸ Walker, J. C. G. Carbon dioxide on the early earth. *Orig. Life Evol. Biosph.* **16**, 117-127 (1985).
- ⁹ Haqq-Misra, J., Kasting, J. F. & Lee, S. Availability of O₂ and H₂O₂ on Pre-Photosynthetic Earth. *Astrobiology* **11**, 293-302 (2011).
- ¹⁰ Luginina, J., Uzuleņa, J., Posevins, D. & Turks, M. Ring-Opening of Carbamate-Protected Aziridines and Azetidines in Liquid Sulfur Dioxide. *European Journal of Organic Chemistry* **2016**, 1760-1771 (2016).
- ¹¹ Kadoya, S. & Tajika, E. Outer Limits of the Habitable Zones in Terms of Climate Mode and Climate Evolution of Earth-like Planets. *Astrophys. J.* **875**, 7 (2019).
- ¹² Kadoya, S. & Catling, D. C. Constraints on hydrogen levels in the Archean atmosphere based on detrital magnetite. *Geochim. Cosmochim. Acta* **262**, 207-219 (2019).
- ¹³ Som, S. M., Catling, D. C., Harnmeijer, J. P., Polivka, P. M. & Buick, R. Air density 2.7 billion years ago limited to less than twice modern levels by fossil raindrop imprints. *Nature* **484**, 359-362 (2012).
- ¹⁴ Som, S. M. et al. Earth's air pressure 2.7 billion years ago constrained to less than half of modern levels. *Nat. Geosci.* **9**, 448-451 (2016).
- ¹⁵ Gebauer, S. et al. Atmospheric Nitrogen When Life Evolved on Earth. *Astrobiology* **20**, 1413-1426 (2020).
- ¹⁶ Pearce, B. K. D., Tupper, A. S., Pudritz, R. E. & Higgs, P. G. Constraining the Time Interval for the Origin of Life on Earth. *Astrobiology* **18**, 343-364 (2018).
- ¹⁷ Kasting, J. F., Zahnle, K. J., Pinto, J. P. & Young, A. T. Sulfur, ultraviolet radiation, and the early evolution of life. *Orig. Life Evol. Biosph.* **19**, 95-108 (1989).
- ¹⁸ R. M. Hazen, *Am. J. Sci.* **2013**, 313, 807-843.
- ¹⁹ Rode, B. M., Fitz, D. & Jakschitz, T. The first steps of chemical evolution towards the origin of life. *Chem. Biodivers.* **4**, 2674-2702 (2007).
- ²⁰ Bujdák, J., Le Son, H., Yongyai, Y. & Rode, B. M. The effect of reaction conditions on montmorillonite-catalysed peptide formation. *Catal. Lett.* **37**, 267-272 (1996).

REVIEWER COMMENTS

Reviewer #1 (Remarks to the Author):

Thanks for considering the referee replies and the point by point. I am very happy with the replies. Whilst I could pick on small things the manuscript has improved to an even higher quality. I cannot wait for it to be out and for people to debate these ideas. Hence I think it should be published ASAP.

Reviewer #2 (Remarks to the Author):

The authors have adequately addressed all of my concerns and I recommend accepting the revised manuscript in its current form.

Reviewer #3 (Remarks to the Author):

Major Comments

Lines 60, 230-231: The authors continue to assert that liquid SO₂ was “probably” available: “suggest the possibility of liquid SO₂ ponds near volcanoes on the Hadean especially since it would probably have had a higher total atmospheric pressure .”

“In the Hadean eon, this was probably liquid, due to the higher atmospheric pressure”

I cannot emphasize enough that this is an extreme picture of early Earth. For example, examining Figure 3 of the Zahnle et al 2010 paper cited by the authors, note that the schematic pCO₂ drops off to ~0.1 bar by 10-100 million years after the moon-forming impact. During this time the surface temperature was also enormously high, on the order of ~500 K, where according to the phase diagram supplied by the authors liquid SO₂ is not stable. This is a typical problem of invoking high surface pressures: they tend to come with high temperatures due to an enhanced greenhouse effect (see also Figure 1, top panel of the Kasting & Ackerman 1986 also cited by the authors). In other words, these references do not predict simultaneous low temperatures and high pressures on early Earth. Therefore, these references do not justify liquid SO₂ on early Earth, certainly not with the minimal discussion and high confidence language used by the authors.

A more promising venue for liquid SO₂ in this reviewer’s opinion comes from a cold early Earth. I earlier gave Kadoya et al. 2018 as a reference, but this was an error, for which I apologize. The correct reference is Kadoya et al. 2019 (<https://agupubs.onlinelibrary.wiley.com/doi/10.1029/2019GC008734>). I strongly encourage the authors to map out the relatively restricted region of parameter space which allows liquid SO₂ that is also predicted by the early climate models, even within their generous uncertainties.

A side note is that Kasting et al 1989 only mention atmospheric degradation of SO₂ because they do not consider liquid SO₂ to be present. I doubt liquid SO₂ would be photostable. The authors are correct in pointing out the photostability problem applies to a broad range of molecules; it must be addressed for those molecules as well.

Minor comments:

Line 127-132: On what basis is it known that covellite was Hadean? How widespread was it on early Earth?

In exploring SIPF: Can you comment on the implications of high salt concentrations for other parts of the protolife apparatus, like vesicles and nucleic acids.

Lines 130-132: Can the authors please expand on the results. >From what is written, it sounds like covellite is much worse than pure Cu(II) as a catalyst (e.g., no tri- or tetra-peptides). How does the yield compare.

The authors provide good replies to these questions in their response letter, but I did not see these replies reflected in their manuscript text (If I missed them, I request the authors to please point them out to me). Can the authors please add this valuable discussion into the paper as it provides critically important caveats that would otherwise elude the broad reader that Nature Communications targets

List of Changes

Peptide formation as on the early Earth: from amino acid mixtures to peptides in sulphur dioxide

On behalf of all the authors, I would like to thank the competent reviewers for providing us with great feedback on our manuscript. We greatly appreciate all the helpful suggestions and valuable comments provided by the reviewers to improve the quality of the manuscript.

Reviewer #1 (Remarks to the Author):

Thanks for considering the referee replies and the point by point. I am very happy with the replies. Whilst I could pick on small things the manuscript has improved to an even higher quality. I cannot wait for it to be out and for people to debate these ideas. Hence I think it should be published ASAP.

Thank you very much for the encouraging comments!

Reviewer #2 (Remarks to the Author):

The authors have adequately addressed all of my concerns and I recommend accepting the revised manuscript in its current form.

Thank you very much for accepting our revisions.

Reviewer #3 (Remarks to the Author):

Major Comments

Lines 60, 230-231: The authors continue to assert that liquid SO₂ was “probably” available: “suggest the possibility of liquid SO₂ ponds near volcanoes on the Hadean especially since it would probably have had a higher total atmospheric pressure .”

“In the Hadean eon, this was probably liquid, due to the higher atmospheric pressure”

I cannot emphasize enough that this is an extreme picture of early Earth. For example, examining Figure 3 of the Zahnle et al 2010 paper cited by the authors, note that the schematic pCO₂ drops off to ~0.1 bar by 10-100 million years after the moon-forming impact. During this time the surface temperature was also enormously high, on the order of ~500 K, where according to the phase diagram supplied by the authors liquid SO₂ is not stable. This is a typical problem of invoking high surface pressures: they tend to come with high temperatures due to an enhanced greenhouse effect (see also Figure 1, top panel of the Kasting & Ackerman 1986 also cited by the authors). In other words, these references do not predict simultaneous low temperatures and high pressures on early Earth. Therefore, these references do not justify liquid SO₂ on early Earth, certainly not with the minimal discussion and high confidence language used by the authors.

We added a longer discussion to the introduction about the possibility of liquid SO₂. According to the various models, there were ‘short’ phases with higher pressure and moderate temperature, where SO₂ existed in liquid form. We also included models and evidence for the snowball earth. It is also important to mention, that local anomalies exist, where the temperature can be very low.

The question about the reaction conditions at the time of the origin of life is highly complex and geochemical data are not available, because almost no rock samples from the Hadean eon are available and it would be highly difficult to estimate the atmospheric pressure from them. Also, about the prevailing temperature no certain prediction can be made at the present time since it depends substantially on the composition and chemical transformations of the atmosphere. Apart from the general assumptions about the conditions during the emergence of life (~ 200-800 Ma), the possibility of different, location-dependent conditions can be considered. Furthermore, seasons and different microhabitats similar to today's Earth are conceivable. In the recent years, detailed models to simulate the change of the atmospheric and environmental conditions of the early Earth were developed.

Models predict that the probability that the surface temperature of the Earth was less than 273.15 K is 67% at ~200 Ma, thus the climate was cold because of the consumption of CO₂ by ejecta weathering, which could lead to the conditions of a (soft) snowball Earth. There is evidence from samples with negative carbon isotope anomalies in carbonate rocks in Namibia for the existence of a snowball Earth later in the Earth's history, in the Paleoproterozoic (2.5 Ga).

The atmospheric pressure ranges from ~0.01 to 100 bar. It is also assumed that a large amount of the Earth's inventory of nitrogen was in the atmosphere with a partial pressure p(N₂) of ~0.8 bar or even 2-3 bar.

Sulphur dioxide SO₂ has a boiling point of -10°C at 1 bar. The here discussed geochemical models suggest temporal phases and environmental conditions on the early Earth, where SO₂ existed in liquid form. In addition, high concentrations of SO₂ near volcanoes on the Hadean Earth increased the local partial pressure of SO₂.

The following references were added:

Kasting, J. Earth's early atmosphere. Science 259, 920 (1993).

Sleep, N. H. The Hadean-Archaean Environment. *Cold Spring Harb. Perspect. Biol.* 2, a002527 (2010).

Turner, J. et al. Record low surface air temperature at Vostok station, Antarctica. *J. Geophys. Res.* 114, 1-14 (2009).

Zahnle, K., Schaefer, L. & Fegley, B. Earth's earliest atmospheres. *Cold Spring Harb. Perspect. Biol.* 2, a004895 (2010).

Kadoya, S., Krissansen-Totton, J. & Catling, D. C. Probable cold and alkaline surface environment of the hadean earth caused by impact ejecta weathering. *Geochem. Geophys. Geosyst.* 21, e2019GC008734 (2020).

Hoffman, P. F., Kaufman, A. J., Halverson, G. P. & Schrag, D. P. A Neoproterozoic snowball Earth. *Science* 281, 1342-1346 (1998).

Kirschvink, J. L. et al. Paleoproterozoic snowball Earth: extreme climatic and geochemical global change and its biological consequences. *PNAS* 97, 1400 (2000).

Kasting, J. F. & Ackerman, T. P. Climatic consequences of very high carbon dioxide levels in the earth's early atmosphere. *Science* 234, 1383-1385 (1986).

Zahnle, K. et al. Emergence of a habitable planet. *Space Sci. Rev.* 129, 35-78 (2007).

Bergstrom, F. W. The Boiling Points of Ammonia, Sulfur Dioxide and Nitrous Oxide. *J. Phys. Chem.* 26, 876-894 (1922).

Bergstrom, F. W. The Vapor Pressure of Sulfur Dioxide and Ammonia. *J. Phys. Chem.* 26, 358-376 (1922).

A more promising venue for liquid SO₂ in this reviewer's opinion comes from a cold early Earth. I earlier gave Kadoya et al. 2018 as a reference, but this was an error, for which I apologize. The correct reference is Kadoya et al. 2019

Thank you for referring to this reference. We corrected this citation and added these findings to the discussion. This publication appeared as Early View in 2019 and was finally published in 2020.

I strongly encourage the authors to map out the relatively restricted region of parameter space which allows liquid SO₂ that is also predicted by the early climate models, even within their generous uncertainties.

We discussed this in the context of the geochemical models.

We added in the discussion section the following sentence, referring to the introduction:

On the early Earth, SO₂ was released by volcanic emissions. As discussed in the introduction, geochemical models suggest temporal phases where SO₂ existed as a liquid. Therefore, SO₂ is an attractive surrogate solvent for prebiotic chemistry.

A side note is that Kasting et al 1989 only mention atmospheric degradation of SO₂ because they do not consider liquid SO₂ to be present. I doubt liquid SO₂ would be photostable. The authors are correct in pointing out the photostability problem applies to a broad range of molecules; it must be addressed for those molecules as well.

SO₂ is a stable gas or liquid. It is photostable. Degradation typically involves oxidation.

Minor comments:

Line 127-132: On what basis is it known that covellite was Hadean? How widespread was it on early Earth?

According to R.M. Hazen, *Am. J. Science* 313, 807 (2013), about 420 different rock-forming or accessory mineral species were widely distributed on Earth and are recognized minerals by the International Mineralogical Association (IMA). It is believed that covellite (CuS) formed under hydrothermal conditions and was widely distributed. It is interesting to note, that the formation of Cu-S bonds in chalcocite (Cu₂S) and covellite (CuS) is dependent primarily on sulphur fugacity f_{S_2} ; at $\log f_{S_2} > -44$ these copper sulfide phases will form at any oxygen fugacity below $\log f_{O_2} < -52$, and thus must be included in an inventory of Hadean minerals. Furthermore, covellite (CuS) belongs to the only plausible copper minerals prior to significant near-surface oxidation.

We therefore added the following sentence to explain this in the manuscript text: Covellite is one of the 13 copper minerals which may have been present in the Hadean and constituted 2.1% of the prebiotic mineral inventory²⁴. It is assumed that covellite was formed by hydrothermal alteration and is one of the plausible copper minerals prior to significant near-surface oxidation²⁴.

In exploring SIPF: Can you comment on the implications of high salt concentrations for other parts of the protolife apparatus, like vesicles and nucleic acids.

Indeed, salt concentrations and the salt composition play an important role in the formation of membranes and protocells. Furthermore, salts can control the activity for example of RNA.

Therefore we added the following sentence and added 3 references:

Recent studies indicate that the salt concentration, altered by wet and dry cycles, might have supported the formation of cell membranes leading to protocells. Furthermore, the salt composition can regulate for example the activity of RNA such as self-replication and extension.

Damer, B. & Deamer, D. The hot spring hypothesis for an origin of life. *Astrobiology* 20, 429-452 (2020).

Mulkidjanian, A. Y., Bychkov, A. Y., Dibrova, D. V., Galperin, M. Y. & Koonin, E. V. Origin of first cells at terrestrial, anoxic geothermal fields. *Proc. Natl. Acad. Sci. USA* 109, E821 (2012).

Matreux, T. et al. Heat flows in rock cracks naturally optimize salt compositions for ribozymes. *Nature Chemistry* 13, doi.org/10.1038/s41557-021-00772-5 (2021).

At the end of the discussion, we rephrased therefore the following sentence:

Although separate environments for the emergence of the first biomolecules and the first organisms are conceivable, high metal concentrations do not seem very favourable for the first organisms. In this study it could be shown that peptides are formed in SO₂ under strongly reduced salt and metal concentrations compare to the SIPF reaction in water. CuCl₂ could be used in catalytic amounts and other additives like NaCl were not necessary at all.

Lines 130-132: Can the authors please expand on the results. >From what is written, it sounds like covellite is much worse than pure Cu(II) as a catalyst (e.g., no tri- or tetra-peptides). How does the yield compare.

Because of the low solubility of covellite, the peptide yields are decreasing to approximately 1/10 of the yield obtained with the pure Cu(II) catalysts. Therefore, we added the following sentence: The amounts of dipeptides formed were about one tenth of those obtained with synthetic CuCl₂ as catalyst.

The authors provide good replies to these questions in their response letter, but I did not see these replies reflected in their manuscript text (If I missed them, I request the authors to please point them out to me). Can the authors please add this valuable discussion into the paper as it provides critically important caveats that would otherwise elude the broad reader that Nature Communications targets.

Thank you very much for this comment. We tried to answer all your questions in our response letter. Now we included these discussions in the manuscript as indicated in this response letter. We highlighted our changes throughout the text.

REVIEWERS' COMMENTS

Reviewer #3 (Remarks to the Author):

The authors have substantially addressed almost all of the concerns I have raised. My sole remaining comment remains regarding the language regarding the prevalence of conditions in which liquid SO₂ would be stable on early Earth. I agree with what the authors write in their response letter, which is that current predictions are that (1) transient conditions compatible with liquid SO₂ may have existed on early Earth, (2) local environments on early Earth might have featured liquid SO₂ for longer intervals, and (3) in a limited number of models, liquid SO₂ might have been available globally on early Earth for geologically long times (Kadoya et al). Therefore, it is worthwhile evaluating liquid SO₂ as a prebiotic solvent.

However, that is not well-communicated by the present language. At present, the authors use the language:

"The here discussed geochemical models suggest temporal phases and environmental conditions on the early Earth, where SO₂ existed in liquid form"

"As discussed in the introduction, geochemical models suggest temporal phases where SO₂ existed as a liquid."

This does not communicate the extreme uncertainty regarding whether liquid SO₂ was present on early Earth. The text should be modified to make this clear. Example modifications are below:

"The here discussed geochemical models suggest the possibility of temporal phases or local environments on the early Earth, where SO₂ existed in liquid form"

"As discussed in the introduction, geochemical models suggest the possibility of temporal phases or local environments where SO₂ existed as a liquid. This motivates consideration of liquid SO₂ as a prebiotic solvent"

List of Changes

From amino acid mixtures to peptides in liquid sulphur dioxide on early Earth

On behalf of all the authors, I would like to thank the competent reviewers for providing us with great feedback on our manuscript. We greatly appreciate all the helpful suggestions and valuable comments provided by the reviewers to improve the quality of the manuscript.

Reviewer #3 (Remarks to the Author):

The authors have substantially addressed almost all of the concerns I have raised. My sole remaining comment remains regarding the language regarding the prevalence of conditions in which liquid SO₂ would be stable on early Earth. I agree with what the authors write in their response letter, which is that current predictions are that (1) transient conditions compatible with liquid SO₂ may have existed on early Earth, (2) local environments on early Earth might have featured liquid SO₂ for longer intervals, and (3) in a limited number of models, liquid SO₂ might have been available globally on early Earth for geologically long times (Kadoya et al). Therefore, it is worthwhile evaluating liquid SO₂ as a prebiotic solvent.

However, that is not well-communicated by the present language. At present, the authors use the language:

“The here discussed geochemical models suggest temporal phases and environmental conditions on the early Earth, where SO₂ existed in liquid form”

“As discussed in the introduction, geochemical models suggest temporal phases where SO₂ existed as a liquid.”

This does not communicate the extreme uncertainty regarding whether liquid SO₂ was present on early Earth. The text should be modified to make this clear. Example modifications are below:

“The here discussed geochemical models suggest the possibility of temporal phases or local environments on the early Earth, where SO₂ existed in liquid form”

“As discussed in the introduction, geochemical models suggest the possibility of temporal phases or local environments where SO₂ existed as a liquid. This motivates consideration of liquid SO₂ as a prebiotic solvent”

Thank you very much for accepting our revisions. We agree and modified the manuscript according to your suggestions. The changes are highlighted.